# Diagnostic Approach and Differences between Spinal Infections and Tumors

**DOI:** 10.3390/diagnostics13172737

**Published:** 2023-08-23

**Authors:** Domenico Compagnone, Riccardo Cecchinato, Andrea Pezzi, Francesco Langella, Marco Damilano, Andrea Redaelli, Daniele Vanni, Claudio Lamartina, Pedro Berjano, Stefano Boriani

**Affiliations:** 1IRCCS Ospedale Galeazzi—Sant’Ambrogio, 20157 Milan, Italy; 2Residency Program in Orthopaedics and Traumatology, University of Milan, 20141 Milan, Italy

**Keywords:** spine infection, spine bone tumor, radiological differential diagnosis, radiological diagnostic

## Abstract

Study design: A systematic review of the literature about differential diagnosis between spine infection and bone tumors of the spine. Background and Purpose: The differential diagnosis between spine infection and bone tumors of the spine can be misled by the prevalence of one of the conditions over the other in different areas of the world. A review of the existing literature on suggestive or even pathognomonic imaging aspects of both can be very useful for correctly orientating the diagnosis and deciding the most appropriate area for biopsy. The purpose of our study is to identify which imaging technique is the most reliable to suggest the diagnosis between spine infection and spine bone tumor. Methods: A primary search on Medline through PubMed distribution was made. We identified five main groups: tuberculous, atypical spinal tuberculosis, pyogenic spondylitis, and neoplastic (primitive and metastatic). For each group, we evaluated the commonest localization, characteristics at CT, CT perfusion, MRI, MRI with Gadolinium, MRI diffusion (DWI) and, in the end, the main features for each group. Results: A total of 602 studies were identified through the database search and a screening by titles and abstracts was performed. After applying inclusion and exclusion criteria, 34 articles were excluded and a total of 22 full-text articles were assessed for eligibility. For each article, the role of CT-scan, CT-perfusion, MRI, MRI with Gadolinium and MRI diffusion (DWI) in distinguishing the most reliable features to suggest the diagnosis of spine infection versus bone tumor/metastasis was collected. Conclusion: Definitive differential diagnosis between infection and tumor requires biopsy and culture. The sensitivity and specificity of percutaneous biopsy are 72% and 94%, respectively. Imaging studies can be added to address the diagnosis, but a multidisciplinary discussion with radiologists and nuclear medicine specialists is mandatory.

## 1. Introduction

Spine infection is a rare disease with different etiological causes and variable outcomes. Its incidence is variable in different areas of the world, accounting for 2–7% of all cases of musculoskeletal infections. Spinal infections can occur in adult and pediatric populations and can involve any component of the vertebral body, intervertebral disc and posterior elements. Extension into the para-spinal soft tissues, the epidural space and the spinal canal can be observed. In case the intervertebral disc is involved, the correct term for the disease is spondylodiscitis, while cases where the vertebra is mainly affected are better described by the terms vertebral osteomyelitis or spondylitis. Spinal infections can threaten patients’ lives but can also lead to intervertebral disc disruption, segmental instability and spinal deformities, reducing the quality of life of affected subjects [1].

Even if spinal infections have been historically reported for hundreds of years, the first historical description of a spinal abscess caused by a tuberculosis infection was made by Pott in 1779. 

Spine infections occur by three major agents: bacteria, causing pyogenic infections, tuberculosis or fungi, responsible for granulomatosis infections, or by parasites, which are the less common etiology [2]. Nowadays, the majority of spinal infections are bacterial monomicrobial [3,4] caused by Staphylococcus aureus in 30 to 80% of cases [5,6]. 

Incidence varies between 1:100,000 and 1:250,000 in developed countries and its estimated mortality rate ranges between 2 and 4% [7,8].

The low specificity of patients’ symptoms and of the radiological signs can delay a correct diagnosis, leading to a late proper therapy. This can often significantly impact the outcomes in a negative way. This delay has been reported in the literature as varying from 2 to 6 months [2] and different proposals can be found in the scientific literature to help physicians to reach a diagnosis. Therapy of spine infection can be variable and multimodal, depending on the extent of the infection, its location in the spine, eventual extra-osseous abscesses, etiology of the infection and eventual presence of hardware. A multidisciplinary approach that involves spine surgeons and infectivologists is frequently recommended. 

On the other hand, incidence of primary spinal tumors (PSTs) is around 2.5–8.5 per 100,000 per year. The rarity of this disease is associated to delayed diagnoses or misdiagnoses [9]. One of the most dangerous errors during the diagnostic process can lead to misdiagnosis of a metastasis for a PST, the spine being a rare site of primary bone tumors and, conversely, being frequently affected by metastatic lesions. Usually, the primary tumors involving the spine are bone tumors of the vertebrae, tumors of the surrounding soft tissues or tumors of the nervous system, as Schwannomas or neurinomas. On the other hand, metastatic disease coming from different organs can affect the spine, with different degrees of local bone rearrangement and consequent exposure to local complications as vertebral fracture or spinal cord compression [10,11,12,13].

Occurrence of PST has some peculiarities: benign primary tumors are more frequent in the younger and frequently involve the posterior elements, while primary malignant tumors occur mostly, but not exclusively, in adults and the elderly and more frequently involve the anterior column [14,15,16,17,18]. Conversely, the spine is the most common site for bone metastases and the incidence is increasing [19,20] because of the increasing age of the population and longer life expectancy due to improvement in cancer oncological medical and radiotherapeutic treatments [9,21,22,23]. It is calculated that spine bone metastases are more frequent than primary malignant spine bone tumors by a factor of 400 to 500 to 1 [24,25].

A diagnosis of PST or spinal metastasis has obviously a huge impact. While malignant tumors can spread with metastases and damage distant organs, benign tumors can be locally aggressive, eventually leading to spinal cord compression and to high rates or local recurrence. This obviously affects the quality of life and the life length expectancy of patients. A fast and proper diagnosis can change the fate of these patients. However, the differential diagnosis between spinal tumors and infections can present different problems in daily practice and can further be influenced by the prevalence of one of the conditions over the other in different areas of the world [26,27,28,29]. Provided that, the definitive diagnosis between infection and tumor must always rely on trocar image-guided biopsy. The histological evaluation should be performed in a high-volume center with expertise in musculoskeletal tumor care.

A review of the existing literature on suggestive or even pathognomonic imaging aspects of both can be very useful for correctly orientating the diagnosis and deciding the most appropriate area for biopsy.

The present paper, addressed to spine surgeons from spine surgeons, has the aim of helping to identify which imaging technique is the most reliable to suggest the diagnosis between spine infection and spine bone tumor. In addition, we tried to clarify which MRI and CT-scan technique is most adequate and which findings are the most useful in the differential diagnosis. 

The major role of radiologists in interpretating imaging must be remembered and respected; however, in the perspective of a multidisciplinary approach, the spine surgeon to whom the patient addresses and that will be responsible for the treatment should be aware of basic knowledge of specific patterns of each disease and should be able to understand which could be the best imaging to orientate the diagnosis. The result must bring the best collaboration with the radiologist, responsible for a correct image interpretation.

## 2. Material and Methods

### 2.1. Study Design

This review aims to investigate the most adequate radiological technique in order to differentiate spinal infections and spinal tumors and, for each technique, to find most specific features for each clinical entity. The PRISMA guidelines were followed during this review’s design, search and reporting stages [30,31,32,33,34,35,36,37]. 

### 2.2. Information Sources and Search Strategy

A primary search on Medline through PubMed distribution used the following search terms: 

(“Spine” [mh]) AND (bone tumor [tiab] OR bone tumors [tiab] OR tumour [tiab] OR bone tumours [tiab] OR bone cancer [tiab] OR bone neoplasm [tiab] OR bone neoplasms [tiab] OR bone neoplastic [tiab] OR bone neoplasia [tiab] OR bone metastasis [tiab] OR bone metastases [tiab] OR metastatic [tiab] OR metastatized [tiab] OR metastatised [tiab] OR discitis [tiab] OR discitis [tiab] OR pyogenic spondylitis or tuberculous spondylitis OR spinal infection) AND (Imaging [tiab] OR diagnosis [tiab] OR MRI [tiab] OR magnetic resonance imaging OR CT scan [tiab] OR computed tomography OR positron emission tomography OR (PET)/CT OR Contrast-enhanced magnetic resonance OR differential diagnosis.)

A filter for English language was applied. Papers were initially identified based on title and abstract. Full-text copies of relevant papers were then obtained and independently evaluated by two reviewers (D.C. and A.P.). When a disagreement between reviewers occurred, it was resolved by a meeting held in consultation with a third author (R.C.). 

Duplicates, reviews, expert’s comments, congress abstracts and articles not in English were excluded. References from the identified articles were checked not to miss any further relevant articles.

### 2.3. Inclusion Criteria

Criteria for inclusion were as follows: -Papers in English;-Papers including patients older than 18;-Papers providing qualitative results on radiological differential diagnosis between spinal tumors and spinal infections, focusing on findings that are useful in CT scan and MRI;-Retrospective or prospective studies including randomized controlled trials, non-randomized trials, cohort studies, case–control studies, case series and other reviews of the literature.

### 2.4. Exclusion Criteria 

Criteria for exclusion were as follows: Articles that did not provide clear results on radiological differential diagnosis between spinal tumors and spinal infections;Articles with patients 18 or younger.

### 2.5. Variables Evaluated 

We identified 5 main groups: Tuberculous;Atypical spinal tuberculosis;Pyogenic spondylitis;Neoplastic (primitive);Neoplastic (metastatic).

For each group, we evaluated the commonest localization, characteristics at CT, CT perfusion, MRI, MRI with Gadolinium, MRI diffusion (DWI) and, in the end, the main features for each group. The level of evidence (LOE) of a given study was assigned based on the scoring system adopted by the North American Spine Society in 2005. The categorization of the studies according to LOE was based on the combined evaluation of two reviewers. When a disagreement between reviewers occurred, it was resolved by a discussion with another author.

## 3. Results

We identified 602 studies through the database search and a screening by titles and abstracts was performed. Based on this, 56 full-text articles were selected and assessed to verify their eligibility for inclusion in the present review. After applying inclusion and exclusion criteria, 34 articles were excluded (Figure 1) and a total of 22 full-text articles were assessed for eligibility (Appendix A). 

### 3.1. Characteristics of Included Studies

Out of the 22 records included, 7 focused their research on radiological differential diagnosis between atypical spinal tuberculosis, tuberculous and pyogenic spondylitis (6 LOE V and 1 LOE I) and the other 15 (2 LOE III and 13 LOE V) focused on the differential diagnosis between different type of spondylitis and neoplastic/metastasis. 

### 3.2. Results of Syntheses

We collected, from the 22 articles included in our study, the role of CT-scan, CT-perfusion, MRI, MRI with Gadolinium and MRI diffusion (DWI) in distinguishing the most reliable features to suggest the diagnosis of spine infection (Table 1) versus bone tumor/metastasis (Table 2).

## 4. Discussion

Infectious diseases of the spine have recently increased their prevalence, even in countries where were considered eradicated, and some of the radiological features of infection can mislead the observer. This group of diseases should not be ignored today in the differential diagnosis with bone tumors of the spine; differentiating spinal tumors and spinal infections is a critical task in clinical practice; it is estimated that 50% of cases of infections are initially confused with tumor [38,39,40]. 

Histologically, acute infection is suppurative and not contained; sub-acute infection is suppurative and contained (abscess), whereas chronic infection is variably suppurative and associated with healing bone remodeling [41]; for this reason, the stage of the disease should be kept in mind in the differential diagnosis.

The literature suggests that the involvement and erosion of the vertebral endplates, with possible disruption of the architecture of the vertebral body, is quite typical of spinal infections; the disease can be extended to multiple segments. Abscesses, in some cases calcified, and inflammatory exudate can be seen in the epidural space and/or in the paraspinal soft tissues [42]. Posterior vertebral elements are normally spared. On MR imaging, a well-defined paraspinal region with abnormal signal intensity; a thin, smooth abscess wall; subligamentous spread to three or more vertebral levels; and multiple vertebral involvement are more suggestive of tuberculous spondylitis than of pyogenic spondylitis. Bone fragments in the intra- and/or extra-spinal soft tissues have been described as characteristic of spinal tuberculosis together with gibbous deformity and severe vertebral collapse, even if these features are most evident in the later stages, when the surgeon has to manage the outcomes [43,44,45].

In atypical pyogenic spondylitis, the involvement of posterior elements, the presence of skip lesions that affect different vertebrae far from each other, and extradural spinal cord compression without bony involvement could be highlighted [46]. 

Single vertebral body lesions (with possible extension to the posterior element), with preserved discs, are more suggestive for spinal tumor. On MR images, the presence of a focus of high signal intensity in the center of an osseous lesion (bull’s-eye) is a negative discriminator for metastasis and a rim of high signal intensity on T2-weighted images around an osseous lesion (halo) is a positive discriminator [47]. On MR images, the diffuse reactive marrow edema in tuberculous and pyogenic infections may simulate hematologic malignancies and metastases [48]. Some cases are discussed in Figure 2, Figure 3, Figure 4 and Figure 5 in order to understand many aspects that could help in the differential diagnosis between infections and tumors.

### 4.1. Differentiating Features on X-ray

Radiological examinations play an important role in this process, and X-rays are often one of the initial imaging modalities employed. Even if X-rays may have limitations in detecting soft tissue details, they can provide valuable information regarding osseous structures and spinal alignment, aiding in the differential diagnosis [29,49]. The most important aspects to consider in the radiographs are the assessment of osseous destruction and bone remodeling patterns:-Tumors of the spine, mainly malignant tumors, can lead to cortical destruction with vertebral collapse and pathological fractures, which are visualized as lytic lesions, loss of trabecular pattern or cortical thinning on X-ray. In addition, periosteal reactions can be shown because of the aggressive bone remodeling, with the formation of Codman’s triangles [50,51,52].-Spinal infections usually cause destruction of the vertebral bodies as well, resulting in a characteristic pattern of vertebral collapse. On X-ray, unlike tumors, it may appear as a “wedge” or “fish-mouth” deformity. Irregular endplates and disc space involvement are additional indicators of an etiology of infection [53,54].

Although it does not represent the exam of choice for evaluating soft tissue, X-rays can provide indirect evidence of abnormalities and paraspinal involvement, aiding in differential diagnosis:-Soft tissue masses or calcifications next to the spine may indicate tumor extension into the paraspinal space, allowing us to be able to hypothesize a spinal tumor etiology. Neural foraminal widening or enlargement may also be observed on X-ray, due to the presence of soft tissue masses eroding bone and compressing adjacent neural structures [55].-In spinal infections, soft tissue swelling, abscess formation or involvement of the psoas muscles may be very common, which may be visible as increased density or widening of the soft tissues on X-ray. The presence of gas within the soft tissues, caused by gas-forming organisms, detected as air lucencies or pneumo-mediastinum, or fistulous communication may be suggestive for an infectious etiology [56].

Lastly, the evaluation of spinal alignment and stability of the segments of the spine is crucial in order to differentiate spinal tumors from spinal infections using X-ray imaging. Deviations from normal spinal alignment can provide valuable clues to the underlying pathology.

-Spinal tumors may clinically manifest as vertebral fractures, listhesis or spinal deformity, such as scoliosis or sagittal or coronal imbalance, which can be shown in an X-ray. Tumor-induced instability may present as spinal subluxation, spondylolisthesis or lateral vertebral body translation. In addition, tumor-induced fractures or destruction can lead to the loss of normal spinal alignment and be the first step in the development of a deformity [55].-Spinal infections can cause vertebral collapse and instability as well, resulting in spinal deformity. However, the deformity associated with spinal infections typically involves a more gradual collapse or wedging of the vertebral bodies, potentially leading to a hyperkyphotic deformity. Furthermore, the presence of associated disc space narrowing with the preservation of adjacent vertebral heights may suggest an infectious etiology [49,57].

Even if X-ray imaging has a role in the diagnosis of infection and tumor and in the differential diagnosis between them, it has many limitations; it has limited sensitivity in detecting early osseous changes, especially in the case of small or lytic lesions. In addition, X-rays may not provide a proper analysis of the soft tissues, making it challenging to state the whole extension of tumor or infection involvement. The literature suggests additional imaging modalities, such as computed tomography (CT) or magnetic resonance imaging (MRI), in order to be able to perform a more comprehensive evaluation [52,58]. Summing up, osseous destruction, bone remodeling patterns, soft tissue abnormalities, paraspinal involvement and spinal alignment are important features to consider in the X-ray images. However, the limitations of X-rays should be taken into account and additional imaging modalities have to be considered with the aim to arrive to an accurate diagnosis, which offer detailed anatomical information and can help differentiate between the two conditions based on specific imaging features.

### 4.2. Differentiating Features on CT

Notwithstanding the usefulness of X-ray in providing valuable information in the differential diagnosis between spinal tumors and spinal infections, additional imaging modalities such as computed tomography (CT) are mandatory in order to further evaluate the extent of the disease and provide better visualization of soft tissues. According to the power of the CT imaging in evaluating the osseous structures of the spine, it may provide detailed cross-sectional images allowing a precise assessment of bone destruction, cortical integrity and showing the presence of osteoblastic or osteolytic lesions. For these reasons, CT scan is reported as the first step in order to orientate the diagnosis: the classic findings of tuberculous spondylitis are large calcified paraspinal soft-tissue masses with thick, irregular rim enhancement with anterior vertebral body destruction [59]. In the spinal pyogenic osteomyelitis, there is fragmented, Swiss-cheese-like, diffuse bony destruction (commonly in the subchondral bone region), associated with paravertebral soft-tissue component with the involvement of the nearby epidural space, together with the destruction of the intervertebral disc with narrowing of the interspace [60,61].

Conversely, bone metastases arise central in the vertebral body with very late involvement of the discs. Soft-tissue involvement is relatively frequent but very rarely expanding inside the psoas fascia as with tuberculous infections.

Osteoblastic involvement is a typical sign of neoplasia; the distinction between marginal sclerosis, typical of tubercular infection, and osteoblastic involvement could be misleading, and it could be solved considering the first scenario as a thin rim of solid and destroyed bone, while the osteoblastic reaction creates a bone with an increased density, but the trabecular pattern remains identifiable. In a variable number of cases, the diagnosis is not so clear through the evaluation of the CT, so other methods have been developed to clarify cases of difficult interpretation. In the literature, many studies have been found about the usefulness of CT perfusion in differentiation between neoplastic and tuberculous disease of the spine [62]; CT perfusion is not a new technique for the diagnostic field. It has been in place now for over two decades. It is known that dynamic contrast perfusion study has been used to obtain the blood flow map on the basis of the density changes related to passage of contrast material through the tissue [63]. High grades of tumors typically have higher relative blood volume, and that is thought to be due to the higher capillary density resulting from angiogenesis [61]. On the other hand, the inflammatory diseases show low relative blood volume due to lower capillary density. Indeed, blood flow and blood volume in cases of tuberculous lesions would be decreased compared to neoplastic lesions, owing to the difference in the capillary density in the two entities. This is based on the fact that tuberculous lesions are associated with some amount of vasculitic changes, which will reduce the blood flow and volume to the lesion. Shankar et al. reported that, at a relative blood flow (rBF) value of 4, the sensitivity to diagnose inflammatory and neoplastic diseases was 100%, so that an rBF < 4 is very suggestive for spinal tuberculosis. In addition, at a relative blood volume (rBV) of 3.5, the sensitivity to diagnose inflammatory diseases was 100% and that for neoplastic disease was 95%, which means that an rBV < 3.5 is highly indicative of infection [62]. Lehman et al. described that the 99mTc-MDP SPECT/CT could help in differential diagnosis considering the localization of the findings: discitis and paraspinal soft-tissue activity were identified in case of infection, while facet joint and pars interarticularis activity were more common in malignancy [64]. 

18FDGPET Scan can be useful in differential diagnosis. The maximum standard uptake value (SUV) is, in most of the cases, much higher in infectious disease (roughly 15 to 20 and more) compared to benign bone tumors (lower than 5), malignant tumors (5 to 10–12) and hematogenous tumors (10 to 18) [65]. Lastly, spinal tumors usually exhibit heterogeneous enhancement after the administration of the contrast on CT imaging, which indicates increased vascularity. Instead, spinal infections may show irregular enhancement, often indicating the presence of an abscess or granulation tissue [53,66].

### 4.3. Differentiating Features on MRI

In comparison to the other examination, the scientific literature underlines that MRI is most sensitive and the imaging modality of choice to determine the spread of disease to bone and to evaluate soft tissues, including the spinal cord, intervertebral discs and paraspinal structures; in addition, the changes on both T1- and T2-weighted images, due mainly to the increased water content of the inflammatory and associated ischemic changes in the bone marrow, allow early detection of the pathological process. It provides excellent contrast resolution and can help in distinguishing between tumor and infection based on various imaging features. 

Although the appearance is highly variable on MRI, the characteristics that most strongly correlate with tumoral lesions are cortical invasion that appears undefined; they appear as well-defined soft tissue masses with different signal intensity on MRI. They may show hypointensity on T1-weighted images and hyperintensity on T2-weighted and STIR images and they often exhibit avid enhancement following contrast administration (in MRI with Gadolinium, a diffusely enhancing is usually highlighted [36]). The sclerotic lesions are hypointense on all the sequences. 

Notwithstanding the epidural extension and the neural structure involvement, which are the main features spondylodiscitis [6,67] spinal infections can present with different MRI findings depending on the stage and severity of the infection, the literature reports three different stages in the evolution of a vertebral infectious condition, which are described by different frameworks in MRI:-STAGE 1 => subtle changes, such as increased T2*-weighted signal intensity within the vertebral body, may be evident, indicating marrow edema or inflammation.-STAGE 2 => T1*-weighted: low signal; T2*-weighted: high signal in the bone marrow and intervertebral disc with heterogeneous enhancement may be observed, reflecting the presence of necrosis, abscess formation or granulation tissue [68,69]; the epidural abscesses show peripheral enhancement with central non-enhancing component.-STAGE 3 => T1*- and T2*-weighted low signal is due to vertebral collapse and endplate sclerosis.

In addition, a low signal on T2*-weighted in the intra- and/or extraspinal soft tissues may highlight small bone fragments and calcification, that has been described as characteristic of spinal tuberculosis and has not been recorded to date in patients with other spinal infections and neoplasms [70,71].

Furthermore, the differences in involvement of adjacent soft tissues are an important differentiating factor between the two entities. 

Spinal tumors typically can extend beyond the vertebral column, infiltrating the paraspinal muscles, neural foramina or even spinal canal, eroding the cortical bone and infiltrating neighboring structures. This extension can be seen as soft tissue masses, neural foraminal widening or spinal canal stenosis on MRI. Additionally, tumor-related epidural compression and spinal cord compression may be observed, resulting in the possibility to analyze the effect of the presence of the tumor indirectly through cord signal changes or cord compression symptoms [55].

In spinal infections, MRI can demonstrate the spread of infection to the adjacent soft tissues, which may manifest itself through paraspinal or epidural abscess formation. These abscesses often manifest as areas of hyperintensity on T2*-weighted images with just a rim of enhancement following contrast administration. MRI can also provide valuable information about the involvement of the spinal cord or nerve roots, such as cord edema, nerve root enhancement, or meningeal enhancement, which are indicative of an infectious process [72].

This examination shows that, according to the overlapping of the features of the two entities, the MRI alone is unreliable in distinguishing necrosis and abscesses from tumor. 

MRI diffusion-weighted imaging (DWI) is an advanced MRI technique that has been reported to be very promising in the differential diagnosis between infections and tumors in recent years; it measures the random movement of water molecules, which is expressed through apparent diffusion coefficient (ACD). Higher ADC values have been reported in spinal infection and malignancy than in normal marrow, because these conditions disrupt bone trabeculae and cause a local increase in water movement. 

However, different studies have reported some differences between infective and tumoral issues: in the acute stage of spinal pyogenic of tubercular infection, it shows restricted diffusion, due to the presence of cellular debris and high viscosity; this restricted diffusion is often observed as hyperintensity on DWI and hypointensity on apparent diffusion coefficient (ADC) maps. In chronic stages, no diffusion restriction with high apparent diffusion coefficient (ADC) values is shown, according to the replacement of marrow fat by inflammatory cells and proteins.

The solid soft tissue mass and the necrotic aspect associated with tumors has restricted diffusion and lower ADC values. However, ADC determination does not allow reliable differentiation of tumoral central necrotic area from abscesses [73].

Summing up, specific imaging features such as enhancement patterns, signal intensities, involvement of adjacent structures and diffusion characteristics can aid in distinguishing between these two conditions. However, a comprehensive approach that considers clinical information is essential for accurate diagnosis and management. It is important to note that imaging findings should always be interpreted in conjunction with clinical history, laboratory results and other diagnostic tests to arrive at an accurate diagnosis. In some cases, further invasive procedures such as biopsy or culture may be necessary for definitive diagnosis [74,75,76].

### 4.4. Differentiating Features on Nuclear Medicine Imaging

Together with the conventional radiological modalities, nuclear medicine imaging techniques can provide valuable information in the differential diagnosis between spinal tumors and spinal infections. Nuclear medicine studies, such as bone scintigraphy and positron emission tomography (PET), offer functional and metabolic information that can aid in distinguishing between these two conditions [77].

Bone scintigraphy, commonly performed using technetium-99m-labeled diphosphonates (Tc-99m MDP), can assess the overall skeletal involvement in patients suspected of having spinal tumors or infections [64].

In spinal tumors, increased radiotracer uptake can be observed in areas of osteoblastic activity, indicating new bone formation and consequent bone remodeling. This increased uptake may be diffuse or focal, corresponding to areas of tumor infiltration or osteoblastic metastases [78]. On the other hand, spinal infections may exhibit increased radiotracer uptake due to local inflammation and increased blood flow in affected regions [79]. The pattern of radiotracer uptake on bone scintigraphy can provide additional clues for differentiating spinal tumors from infections.

Concerning spinal tumors, the increased uptake is typically focal and shows a variable intensity, depending on the aggressiveness of the tumor. Metastatic lesions often have higher uptake than primary tumors.

Spinal infections tend to show more diffuse and homogeneous uptake, due to the involvement of multiple vertebral levels or the presence of multifocal abscesses [80,81,82].

In the last decades, PET imaging with 18F-fluorodeoxyglucose (FDG) has taken on great importance in the evaluation of spinal tumors and infections. FDG-PET provides metabolic information by measuring glucose uptake, which is typically increased in malignant tumors and infectious processes due to their high metabolic activity. 

Spinal tumors, particularly aggressive and high-grade malignancies, tend to exhibit intense FDG uptake on PET imaging according to the high metabolism of the cells, and the increased uptake may even extend beyond the primary tumor site, indicating metastatic involvement. Even if the spinal infections may demonstrate increased FDG uptake on PET imaging in the same way, reflecting the metabolic activity of the inflammatory process, the pattern of uptake in infections may help to differentiate the two different entities, according to the fact that, in infections, the uptake is often more diffuse and multifocal, involving multiple vertebral levels, paraspinal regions and even adjacent soft tissues. Moreover, PET imaging can help to identify the presence of associated patterns, such as abscesses, which may appear as areas of intense FDG uptake with corresponding low-density areas on CT imaging. 

One of the principal advantages of FDG-PET imaging is the ability to detect subtle lesions in the early stages of pathology, or sites of disease activity that may not be evident on other imaging modalities. However, FDG-PET imaging has certainly many limitations in differentiating between malignant tumors and infectious processes, as both can demonstrate increased glucose metabolism. Therefore, additional imaging studies and clinical correlation are often required to establish the underlying etiology [78,79,80,81]. Hybrid imaging techniques such as PET/CT or PET/MRI can provide a combination of functional and anatomical information, enhancing the diagnostic accuracy, and may represent one the most viable options in order to increase the sensibility and the specificity of the exam. The combination of metabolic PET data with the detailed anatomical imaging from CT or MRI can facilitate precise localization of the lesions and aid in the differentiation process [49].

In conclusion, it is worth noting that nuclear medicine imaging techniques should be used judiciously, considering factors such as radiation exposure, cost, availability and patient-specific clinical context. These studies are typically reserved for situations where there is diagnostic uncertainty or a need to evaluate the whole-body extent of disease involvement.

## 5. Conclusions

Definitive differential diagnosis between infection and tumor requires biopsy and culture. The sensitivity and specificity of percutaneous biopsy are 72% and 94%, respectively [27]. The rate of success is also modified if using a trocar or a thin needle. 

Imaging studies can be added to address the diagnosis. Through CT, MRI and PET-scan, surgeons can be supported in analyzing the signs that are typical of the two conditions. A multidisciplinary discussion with radiologists and nuclear medicine specialists is mandatory. 

## Figures and Tables

**Figure 1 diagnostics-13-02737-f001:**
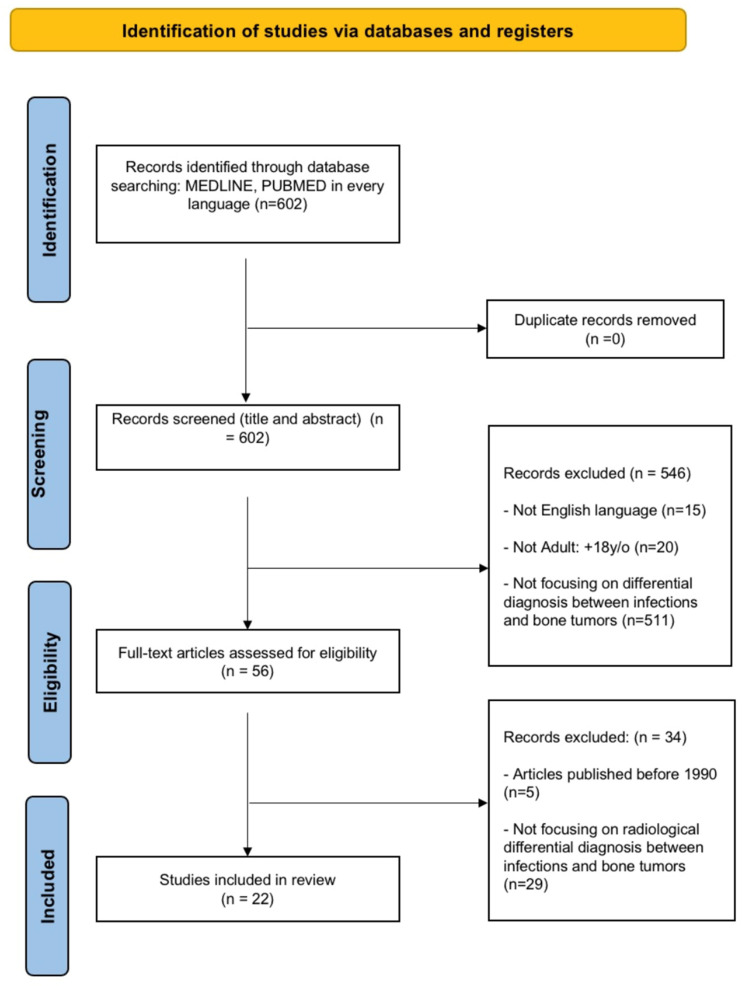
PRISMA 2020 flow diagram for new systematic reviews.

**Figure 2 diagnostics-13-02737-f002:**
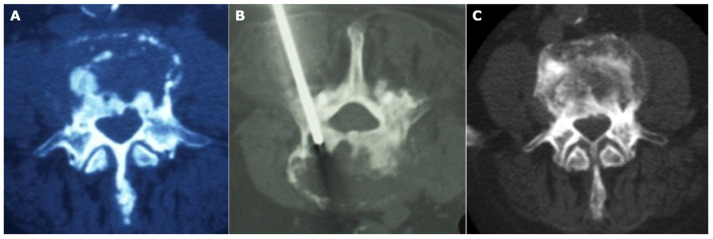
Case of a 77-year-old male with a previous diagnosis of lung tumor 4 years before. At the CT scan (**A**), a lytic lesion was observed, considered as a metastatic lesion from lung cancer. The patient was submitted to radiotherapy with a progression of pain and of bony erosion. After a biopsy was made (**B**), the diagnosis of spondylodiscitis was confirmed and, after an appropriate antibiotic therapy, the local situation healed with a regression of symptoms (**C**).

**Figure 3 diagnostics-13-02737-f003:**
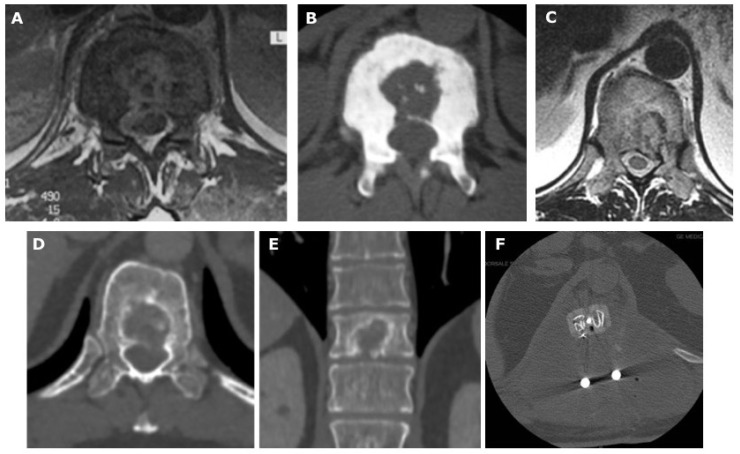
Diagnostic workup of a patient, a 42-year-old male, affected by pyogenic spondylitis. MRI (**A**) and CT scan (**B**) show a lithic lesion of the central portion of the body, with little bony fragments in the middle of the lesion. A biopsy revealed the infective nature of the disease and an antibiotic treatment was administered. On the other side, a 47-year-old man showed a similar lesion on MRI (**C**) and CT scan (**D**,**E**), with a peripheral sclerotic rim and some little islands of bone in the lithic lesion. In this case, the CT-guided biopsy with trocar revealed a central chondrosarcoma, for which an en bloc resection was performed (**F**).

**Figure 4 diagnostics-13-02737-f004:**
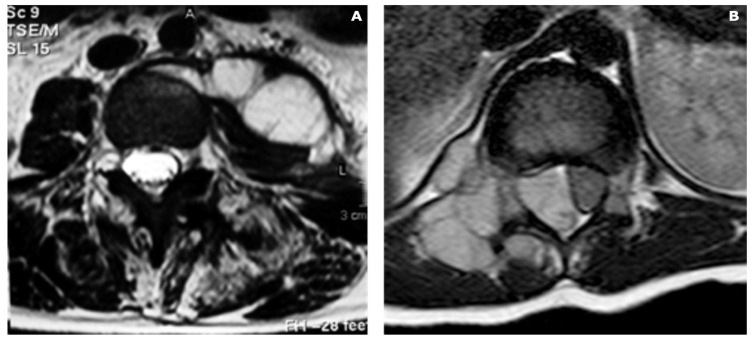
Comparison of an abscess in a spondylodiscitis case (**A**) and an extra-osseous extension of a Ewing sarcoma (**B**). While, in the first case, an evacuation of the abscess followed by antibiotic therapy were performed, in the second case, an adjuvant chemotherapy was mandatory to shrink the tumor mass. Chemotherapy was then followed by an en bloc resection of the spine with a complete excision of the tumor, with margins free of tumor. Note the absence of bony erosion in the first case, while, in the second case, the tumor produced a resorption of the right pedicle.

**Figure 5 diagnostics-13-02737-f005:**
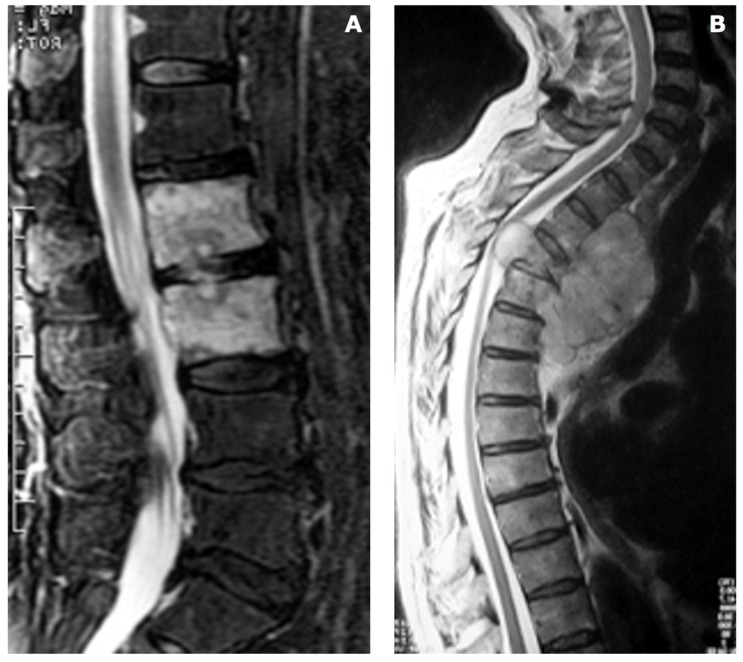
Disc involvement can help in differentiating infections from tumors. In case (**A**), a patient was suffering from bacterial spondylodiscitis (pathogen: Brucella), where the disc between the two infected vertebrae is affected by the infection, as seen in the MRI, where the disc shows a white lesion that connects the two vertebral bodies. In case (**B**), the patient was affected by a multiple metastatic lesion of the thoracic spine. The epidural mass compressing the spinal cord is clearly visible in the MRI scan but significant to helping the diagnosis is the sparing of the discs among the affected vertebrae.

**Table 1 diagnostics-13-02737-t001:** Results of syntheses—infections.

	Commonest Location	X-ray	CT	CT-Perfusion	MRI	MRI with Gadolinium	MRI-Diffusion (DWI)	Features
**Spinal Tuberculosis**	Cervical and Lumbar spine; A single segment is involved	Collapse and destruction of the vertebra Loss of disc space “Gibbus deformity” Anterior wedging or kyphosis	Erosions, marginal sclerosis, and sequestra formation Lytic lesions Epidural and paraspinal involvement Spinal canal involvement Small bony fragment	relative Blood Flow (rBF) value < 4 relative Blood Volume (rBV) value < 3.5	**Early stage**: hypointense on T1-weighted; hyperintense on T2-weighted (marrow edema, inflammation). granulomatous inflammatory response **Chronic stage**: hypointense on T1-weighted and on T2-weighted images (necrosis, abscess formation) Involvement of adjacent structures (spinal cord, nerve roots, and paraspinal tissues)	Heterogeneous enhancement (necrosis, abscess formation, or granulation tissue)	**Early stage**: restricted diffusion (low ADC values) **Chronic stage**: no diffusion restriction with high ADC values	Involvement of vertebral bodies and the related disc with rapid destruction Subligamentous spread with extension into the paraspinal soft tissue with abscesses, usually calcified Extension in epidural space with spinal cord or nerve compression Posterior elements of the vertebra are generally spared
**Pyogenic Spondylitis**	Cervical and Lumbar spine; A single segment is involved	Localized vertebral destruction Loss of disc height Paraspinal soft tissue swelling Bony sequestra, sclerosis, and vertebral collapse	Erosions, sclerosis, and sequestra formation Cortical destruction Sequestration and abscesses	relative Blood Flow (rBF) value < 4 relative Blood Volume (rBV) value < 3.5	**Early stage**: hypointense on T1-weighted; hyperintense on T2-weighted (inflammation). granulomatous inflammatory response **Chronic stage**: hypointense on T1-weighted and on T2-weighted images (necrosis, abscess formation) Involvement of adjacent structures (spinal cord, nerve roots, and paraspinal tissues)	Enhancing inflammatory changes, abscesses, and granulation tissue Ring-enhancing abscesses with peripheral rim enhancement	**Early stage**: restricted diffusion (low ADC values) **Chronic stage**: no diffusion restriction with high ADC values	Involvement of vertebral bodies and the related disc with rapid destruction Subligamentous spread with extension into the paraspinal soft tissue with abscesses, usually calcified Extension in epidural space with spinal cord or nerve compression Posterior elements of the vertebra are generally spared
**Atypical Spinal Tuberculosis**	Thoraco-lumbar spine; Multisegmental	Minimal vertebral body involvement Isolated disc space narrowing Abnormal spinal alignment, or signs of instability	Large paravertebral abscesses		Isolated involvement of the posterior elements or a skip lesion pattern Heterogeneous mixed T2-weighted Epidural extension			Involvement of posterior elements Skip lesion separated from each other Extradural spinal cord compression

**Table 2 diagnostics-13-02737-t002:** Results of syntheses—tumors.

	Commonest Location	X-ray	CT	CT-Perfusion	MRI	MRI with Gadolinium	MRI-Diffusion (DWI)	Features
**Neoplastic (Primitive)**	Mostly Thoracic spine	Vertebral collapse Pathological fractures Abnormal spinal alignment	Destruction, erosion, sclerosis Presence of a bony mass Spinal canal involvement Erosion of the cortical bone Presence of spinal cord compression or nerve root impingement	relative Blood Flow (rBF) value > 4 relative Blood Volume (rBV) value > 3.5	Location, size, extension Extent of spinal cord compression, evaluate nerve root involvement, and identify the presence of cystic or necrotic components within the tumor Various imaging features on MRI	Intense contrast enhancement in homogeneous or ring-like enhancement	Restricted diffusion (low ADC values)	Vertebral body and posterior elements involvement Preserved disc Halo Sign
**Neoplastic (Metastasis)**	Mostly Thoracic spine	Osteolytic, osteoblastic, or mixed lytic/blastic lesions Destructive bone lesions Loss of vertebral body height Pathologic fractures	Destruction, cortical thinning A soft tissue mass	relative Blood Flow (rBF) value > 4 relative Blood Volume (rBV) value > 3.5	Location, size, extension Extent of spinal cord compression, evaluate nerve root involvement, and identify the presence of cystic or necrotic components within the tumor Various imaging features on MRI	Homogeneity, heterogeneity, or rim enhancement, can provide insights into the aggressiveness and vascularity of the metastatic lesions	Restricted diffusion (low ADC values)	Vertebral body and posterior elements involvement Preserved disc

## Data Availability

Not applicable.

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
