# Peer review of "Diagnostic Approach and Differences between Spinal Infections and Tumors"

_diagnostics, 2023, doi:10.3390/diagnostics13172737_

Round 1

Reviewer 1 Report

This article is a systematic review of the literature on the differential diagnosis of spinal infections and spinal bone tumors. This article has guiding significance for the identification of clinical spinal related diseases and provides reference value for the compilation of related textbooks. Overall, this review is well organized and comprehensively described. But we found a serious problem with the article that there is no images. This review is mainly a description of images related to the spine, so we believe that appropriate images are a must.

Author Response

Dear Reviewer, thank you for your positive comments. We completely agree with you on the need of images, and we added them to the manuscript. You can find the changes attached in the new version of the manuscript. 

Thank you again for your time in reviewing our study

Reviewer 2 Report

Authors present a review article on diagnostic approach and differences between spinal infections and bone tumors of the spine from point of view of the spine surgeon. 5 main groups were identified: Tuberculous, Atypical Spinal Tuberculosis, Pyogenic Spondylitis, Neoplastic (Primitive, Metastatic). Total of 22 full articles were analyzed. The authors conclude that biopsy and culture remain the gold standard for the final differentiation. This manuscript lacks a justification of its title - what is the spinal surgeons point of view here? I suggest to search out the literature and add several illustrative cases - either own cases of the authors or cases from case reports, with permission of the authors, and to provide on a practical example a value of the diagnostic methods such as CT or MRI. This is a must. I suggest to provide typical imaging i.e. typical cases of the 5 groups of cases. In that way, this manuscript will fulfill his mission. Include a neuroradiologist as a co-author who can review everything from his/her point of view. 

Acceptable. 

Author Response

Dear Reviewer, thank you very much for your precious comments. We completely agree on the need of images, that we have searched and found in our own database of cases. You will find them embedded in the new version of the manuscript. We also checked again the language and modified the title as you requested.

Hope these changes satisfy your comments.

Thank you again for your efforts in improving our paper.

Round 2

Reviewer 1 Report

It is suggested that this article be accepted in its present form, as it has been revised as requested.

Author Response

Dear Reviewer, thank you very much for your feedback. Our best regards

Reviewer 2 Report

Authors have responded to some of the reviewer remarks. I suggest to re-arrange the figures - that the spine is always correctly oriented, and that you provide MRI and CT scans. Also postoperative. 

Ok. 

Author Response

Dear Reviewer, thank you for your comment.

As you correctly requested we have changed the orientation of the figures in the text.

Regarding the postoperative pictures, we didn't included it initially as the main scope of the paper was diagnostic. However, following your suggestion, we included the postoperative CT scan of the en bloc resection of the chondrosarcoma, where you can appreciate a carbon fiber cage filled with bone graft instead of the vertebral body. The other case was treated conservatively so no postoperative pictures are available. We hope this can satisfy your request.

Our best regards and thanks for the efforts in improving our paper